# DiffuSolve: Diffusion-based Solver for Non-convex Trajectory Optimization

## Abstract

Optimal trajectory design is computationally expensive for nonlinear and high-dimensional dynamical systems. The challenge arises from the non-convex nature of the optimization problem with multiple local optima, which usually requires a global search. Traditional numerical solvers struggle to find diverse solutions efficiently without appropriate initial guesses. In this paper, we introduce `DiffuSolve`, a general diffusion model-based solver for non-convex trajectory optimization. An expressive diffusion model is trained on pre-collected locally optimal solutions and efficiently samples initial guesses, which then warm-starts numerical solvers to fine-tune the feasibility and optimality. We also present `DiffuSolve+`, a novel constrained diffusion model with an additional loss in training that further reduces the problem constraint violations of diffusion samples. Experimental evaluations on three tasks verify the improved robustness, diversity, and a $2\times$ to $11\times$ increase in computational efficiency with our proposed method, which generalizes well to trajectory optimization problems of varying challenges.

## 1 Introduction

Optimal trajectory design is fundamental in decision-making for autonomous agents. In the open-loop case, with a predetermined initial configuration, the goal is to plan an optimal path and control sequence for an agent to reach a target while satisfying the dynamics and safety constraints. In the case of complex nonlinear system dynamics and environmental constraints, these problems often have non-convex structures. For example, within a chaotic dynamical system like the Circular Restricted Three-Body Problem (CR3BP) (Koon et al., 2000), the trajectory optimization problem for the spacecraft has multiple local optima of different qualitative behaviors with various tradeoffs (Hartmann et al., 1998; Russell, 2007). Identifying these diverse, locally optimal, and feasible solutions efficiently remains a significant challenge in the field.

Control transcription of the open-loop trajectory optimization problem involving nonlinear functions results in a nonlinear program (NLP), where an initial guess is required to get the solution (Betts, 1998). When one is interested in solving for several solutions, a two-step global search is needed: (i) sampling a distribution on the control space, and (ii) using this sample as an initial guess to the NLP solver that hopefully generates a solution. For each solution from the NLP, optimality (and also feasibility) is often verified by the solver by checking whether the first-order necessary conditions, the Karush-Kuhn-Tucker (KKT) conditions, are satisfied (Kuhn & Tucker, 2013; Karush, 1939). However, the high dimensionality and non-convex nature of the problem often make the two-step global search process time-consuming such that the efficiency demands of autonomous systems are not met.

Machine learning has also gained popularity in trajectory optimization, but primarily focuses on solving the convexified problem rather than learning the solution distribution to the original non-convex problem. For example, investigations have been made on warm-starting Quadratic Programs (QP) using neural networks (Chen et al., 2022; Cauligi et al., 2021; Sambharya et al., 2023). Recently, generative models, particularly diffusion models, have shown good performance in generating trajectories or controls from a distribution (Janner et al., 2022; Ajay et al., 2022; Chi et al., 2023; Carvalho et al., 2023; Sridhar et al., 2023). When conditioned on the environmental parameters, diffusion models can efficiently generate appropriate and diverse trajectories even in an *a priori* unknown environment. However, the diffusion model alone without the NLP solver may output trajec-

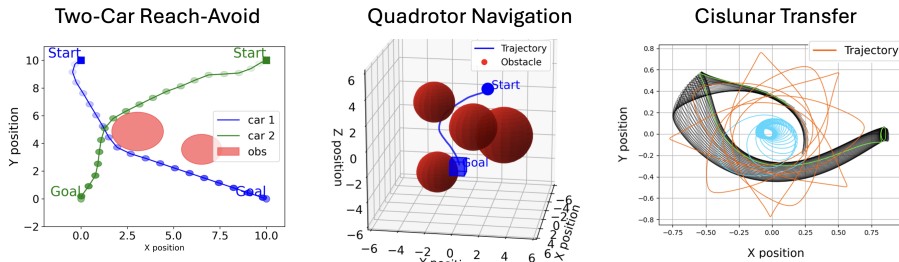

Figure 1: Three trajectory optimization tasks in our experiments (from left to right): *two-car reach-avoid* problem; *quadrotor navigation* problem; *cislunar transfer* problem. The dimensions of the decision variables are 41, 241, and 64, respectively.

tories that violate the dynamic and safety constraints, causing catastrophic damage to safety-critical systems. It also lacks the ability to verify the optimality of the sampled trajectories. While efforts have been made to integrate safety constraints into generative models (Li et al., 2023; Chang et al., 2023; Yang et al., 2023; Xiao et al., 2023; Power et al., 2023; Botteghi et al., 2023), it is still challenging to achieve solution quality, diversity, feasibility, and computational efficiency at the same time.

In this paper, we propose `DiffuSolve`: a diffusion model-based solver for non-convex trajectory optimization with improved efficiency, diversity, and robustness. Our contribution is two-fold: First, we train a conditional diffusion model to learn the complex solution distribution of non-convex optimization problems, and efficiently sample a set of diverse and high-quality trajectory predictions. These samples are then used as warm starts for an NLP solver to derive locally optimal and feasible solutions, ensuring the robustness of `DiffuSolve`. Second, to mitigate the effect of diffusion samples that violate the problem constraints, we developed `DiffuSolve+` with a novel constrained diffusion model that incorporates an additional constraint violation loss in the training process.

`DiffuSolve` has the following appealing properties: **(a). Computational Efficiency.** The diffusion model samples high-quality solutions that are close to local optima as warm starts, from which the NLP solver can quickly converge. **(b). Robustness and Diversity.** A constrained diffusion model produces diverse samples with minimal constraint violations, further refined by an NLP solver for enhanced feasibility and optimality. **(c). Automatic Data Generation.** With an NLP solver, we can generate diverse problems and collect corresponding solutions ourselves, allowing for on-demand data augmentation whenever required. **(d). Generalization.** Our framework is suitable for trajectory optimization across various system dynamics and environment settings, with the potential for optimization problems in other areas like finance (Sambharya et al., 2023), etc.

We demonstrate the efficacy of our method on three non-convex trajectory optimization problems: *two-car reach-avoid*, *quadrotor navigation*, and *cislunar low-thrust transfer* for a spacecraft.

## 2 RELATED WORK

**Machine Learning for Optimization.** Machine learning methods have been widely used to solve a variety of optimization problems with different techniques, including convex Quadratic Programming (QP) (Zeilinger et al., 2011; Zhang et al., 2019; Chen et al., 2022; Sambharya et al., 2023), mixed-integer programming (Cauligi et al., 2021), combinatorial optimization (Sun & Yang, 2023; Huang et al., 2023) and black-box optimization (Kumar & Levine, 2020; Trabucco et al., 2022; Krishnamoorthy et al., 2023), etc. For the Combinatorial Optimization problem, graph-based diffusion models are used to directly generate solutions (Sun & Yang, 2023; Huang et al., 2023). To improve the constraint satisfaction in optimization problems, neural networks have been incorporated with fully differentiable constraint function (Donti et al., 2021). Conditional variational autoencoder (CVAE) and long short-term memory (LSTM) are proposed to warm-start the general nonlinear trajectory optimization problems (Li et al., 2023). Another branch of work leverages machine learning methods and pre-collected solution data points for solving partial differential equations (PDEs) (Cai et al., 2021). One of the most pupular work is physics-informed neural networks (PINN) (Raissi et al., 2019), which introduces the solution prediction model with a hybrid training loss for labeled solution prediction and violation of the original PDEs.

**Diffusion Models.** The diffusion models (Sohl-Dickstein et al., 2015; Song et al., 2020b) is able to sample the multi-modal distribution with a stochastic differential equation (SDE). The models are further developed into denoising diffusion probabilistic models (DDPM) (Ho et al., 2020), denoising diffusion implicit models (DDIM) (Song et al., 2020a), etc., for application of image generation. Both classifier-guided diffusion (Dhariwal & Nichol, 2021) and classifier-free guidance (Ho & Salimans, 2022) are proposed to achieve conditional generation with diffusion models.

**Diffusion-based Trajectory Generation with Constraints.** Diffusion models are becoming popular in robotics for generating controls or trajectories (Ajay et al., 2022; Janner et al., 2022; Liang et al., 2023; Sridhar et al., 2023; Chi et al., 2023; Mishra et al., 2023; Ding et al., 2024). Many efforts have been made to improve the diffusion model for creating safe trajectories for safety-critical systems. Control barrier functions (Xiao et al., 2023; Botteghi et al., 2023) and collision-avoidance kernels (Chang et al., 2023) are employed to guide diffusion models to sample feasible trajectory solutions. To adapt to new constraints in testing, prior work investigated composing diffusion models to improve the generalizability (Power et al., 2023; Yang et al., 2023).

Different from existing work, our paper focuses on high-dimensional, highly nonlinear trajectory optimization problems with many local optima, and aims to find diverse solutions efficiently. Instead of incorporating constraints in the sampling process, our constrained diffusion model in `DiffuSolve+` includes a novel loss in training, which can be combined with the current work to further reduce the violation. Moreover, `DiffuSolve` and `DiffuSolve+` are equipped with an NLP solver that finetunes the diffusion samples until feasibility and optimality are obtained at its best.

## 3 PRELIMINARIES

**Non-Convex Trajectory Optimization.** We consider a general open-loop trajectory optimization problem: assume a known agent's dynamical model, the predefined initial configuration, and the obstacle settings, we aim to plan an optimal trajectory and controls for the agent to reach a target while satisfying the dynamics and safety constraints. With the direct transcription (Betts, 1998), this trajectory optimization problem can be discretized as the following nonlinear program $\mathcal{P}_y$:

$$\mathcal{P}_y := \begin{cases} \min_{x} & J(x;y) \\ s.t., & g_i(x;y) \leqslant 0, \ i = 1, 2, ..., l \\ & h_j(x;y) = 0, \ j = 1, 2, ..., m \end{cases} \tag{1}$$

where $x \in \mathcal{X} \subset \mathbb{R}^n$ is the decision variable to be optimized, and $y \in \mathcal{Y} \subset \mathbb{R}^k$ represents various problem parameters. $J \in C^1(\mathbb{R}^n, \mathbb{R}^k; \mathbb{R})$ is the objective function. Each $h_i \in C^1(\mathbb{R}^n, \mathbb{R}^k; \mathbb{R})$ and $g_j \in C^1(\mathbb{R}^n, \mathbb{R}^k; \mathbb{R})$ are equality and inequality constraint functions, respectively. In trajectory optimization, $x$ might include a sequence of control $u$, the time variable $t$ required to reach the target, etc. Parameters $y$ can include control limits, target and obstacle characteristics such as position and shape, and other relevant factors. The objective function $J$ can be time-to-reach, fuel expenditure, etc. The constraint functions $g$ and $h$ could contain requirements for dynamical models, goal-reaching, obstacle avoidance, etc.

When the problem parameter $y$ is fixed, $\mathcal{P}_y$ in Eq. (1) typically presents a non-convex optimization problem with multiple local optima. To solve this problem $\mathcal{P}_y$, a gradient-based NLP solver $\pi$ can be used. A local optimum (feasible) $x^*$ to $\mathcal{P}_y$ is verified by the solver $\pi$ that satisfies the KKT conditions with no constrained violation. To find different $x^*$, the user first provides multiple initial guesses $x^0$ for the decision variables $x$ to the solver $\pi$, typically drawn from a uniform distribution or some heuristic approach. Then the solver $\pi$ performs an iterative, gradient-based optimization until a local optimum $x^*$ is found. The popular methods to conduct such gradient-based optimization in non-convex problems include the Interior Point Method (IPM) (Wright, 1997) and the Sequential Quadratic Programming (SQP) (Boggs & Tolle, 1995).

**Diffusion Probabilistic Model.** As expressive generative models, the diffusion probabilistic models (Sohl-Dickstein et al., 2015; Ho et al., 2020; Song et al., 2020b) generate samples from random noises through an iterative denoising process. The forward process (*i.e.*, diffusion process) iteratively adds Gaussian noise to sample $x_k$ at the $k$-th step according to the following equation:

$$q(x_{k+1}|x_k) = \mathcal{N}(x_{k+1}; \sqrt{1 - \beta_k}x_k, \beta_k \mathbf{I}), \quad 0 \leqslant k \leqslant K - 1. \tag{2}$$

where $\beta_k$ is a given variance schedule. With the diffusion steps $K \to \infty$, $x_K$ becomes a random Gaussian noise. From above one can write:

$$x_k = \sqrt{\bar{\alpha}_k}x_0 + \sqrt{1 - \bar{\alpha}_k}\varepsilon, \tag{3}$$

where $\bar{\alpha}_k = \prod_{k'=1}^{K}(1 - \beta_{k'})$ and $\varepsilon \sim \mathcal{N}(0, \mathbf{I})$ is a random sample from a standard Gaussian. The data generation process is the reverse denoising process that transforms random noise into data sample:

$$p_\theta(x_{k-1}|x_k) = \mathcal{N}\big(x_{k-1}; \mu_\theta(x_k), \sigma_k \mathbf{I}\big), \quad 1 \leqslant k \leqslant K \tag{4}$$

with initial noisy data sampled from a random Gaussian distribution $x_K \sim \mathcal{N}(0, \mathbf{I})$. $p_\theta$ is the parameterized sampling distribution, and it can be optimized through an alternative $\varepsilon_\theta(x_k, k)$ which is to predict $\varepsilon$ injected for $x_k$, for every diffusion step $k$. For conditional generation modeling, a conditional variable $y$ is added to both processes $q(x_{k+1}|x_k, y)$ and $p_\theta(x_{k-1}|x_k, y)$, respectively. The classifier-free guidance (Ho & Salimans, 2022) can be applied to further promote the conditional information, which learns both conditioned and unconditioned noise predictors as $\varepsilon_\theta(x_k, k, y)$ and $\varepsilon_\theta(x_k, k, \varnothing)$. Specifically, $\varepsilon_\theta$ is optimized with the following loss function:

$$\mathcal{L}_{\mathrm{diff}} = \mathbb{E}_{(x_0, y) \sim \mathcal{X} \times \mathcal{Y}, k, \varepsilon, b} \left\| \varepsilon_\theta\Big(x_k(x_0, \varepsilon), k, (1 - b) \cdot y + b \cdot \varnothing\Big) - \varepsilon \right\|_2^2 \tag{5}$$

where $x_0$ and $y$ are sampled from groundtruth data, $\varepsilon \sim \mathcal{N}(0, \mathbf{I})$, $k \sim \mathrm{Uniform}(1, K)$, $b \sim \mathrm{Bernoulli}(p_{\mathrm{uncond}})$ with given unconditioned probability $p_{\mathrm{uncond}}$, and $x_k$ follows Eq. (3).

## 4 METHODOLOGY

We first propose `DiffuSolve`, a general framework for efficient and robust non-convex trajectory optimization using diffusion models and NLP solvers. Then we introduce a novel constrained diffusion model specifically designed to guide the diffusion models to generate feasible solutions, which is included in `DiffuSolve+`.

### 4.1 DIFFUSOLVE FRAMEWORK

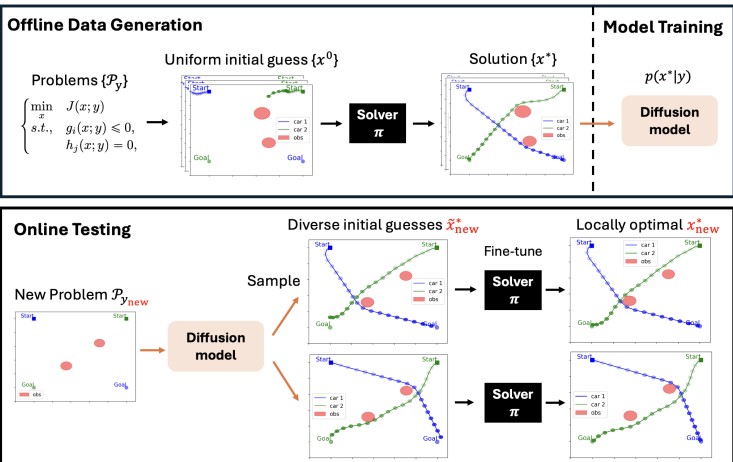

Figure 2: The `DiffuSolve` framework for solving non-convex trajectory optimization problems.

It has been observed that for similar trajectory optimization problems $\mathcal{P}_y$ in Eq. (1), their solutions $x^*$ often exhibit similar structures (Amos et al., 2023; Li et al., 2023). Similar problems are those $\mathcal{P}_y$ with the same objective $J$ and constraints $g, h$, but with different parameters $y$. Thus, the key insight of `DiffuSolve` is to use a diffusion model to learn $p(x^*|y)$, the conditional probability distribution of locally optimal solutions based on pre-solved similar problems. Then the diffusion model can generalize and predict diverse solutions $\tilde{x}^*_{\mathrm{new}}$ to new scenarios where the condition $y_{\mathrm{new}}$ value is unseen. However, diffusion models inevitably make prediction errors, which can result in constraint violations. To address this, we use diffusion samples $\tilde{x}^*_{\mathrm{new}}$ as initial guesses for the NLP solver, which

fine-tunes them to derive the final solutions $x_{\text{new}}^*$, increasing both feasibility and optimality. This approach acts like a safety net, enhancing the robustness of the solutions.

We illustrate our proposed `DiffuSolve` framework in Fig. 2. In the offline data generation process, we use an NLP solver $\pi$ to collect locally optimal and feasible solutions $x^*$ for various problem instances $\mathcal{P}_y$ with uniformly sampled initial guesses. The solutions with sub-optimal objective values are filtered out manually and the remaining dataset is used as the training data. For `DiffuSolve` training, we use a conditional diffusion model to learn $p(x^*|y)$ with classifier-free guidance (Ho & Salimans, 2022), with the loss function in Eq. 5. In the online testing process when a new problem $\mathcal{P}_{y_{\text{new}}}$ is presented with *a priori* unknown value $y_{\text{new}}$, we first use the diffusion model to predict diverse solution candidates $\tilde{x}_{\text{new}}^*$ as initial guesses. Then the NLP solver $\pi$ only requires minor adjustments to derive the final solutions $x_{\text{new}}^*$ with improved feasibility and optimality. This warm-starting approach is fully parallelizable for both diffusion sampling and the solving process of $\pi$.

## 4.2 DIFFUSOLVE+ WITH CONSTRAINED DIFFUSION MODEL

The original diffusion models in `DiffuSolve` aim to learn the distribution of locally optimal solutions $p(x|y), x \in \mathcal{X}, y \in \mathcal{Y}$ with $\mathcal{X}$ as the variable space and $\mathcal{Y}$ as the problem parameter space. `DiffuSolve` is optimized through the loss function as Eq. (5) without knowing any constraint information. As a result, the sampled solutions could be close to the groundtruth but still largely violate the constraints. To address this issue, we propose an improved `DiffuSolve+` method with a constrained diffusion model (CDM) to minimize the constraint violations in the sampled trajectories. Inspired by the equation loss in PINN (Raissi et al., 2019), we define a violation function $V(x, y) : \mathcal{X} \times \mathcal{Y} \rightarrow \mathbb{R}$ for differentiable constraints in Eq. (1):

$$V = \sum_{i=1}^{l} \max(g_i, 0) + \sum_{j=1}^{m} |h_j|, \tag{6}$$

where $V(\cdot, \cdot) \in PC^1(\mathbb{R}^n, \mathbb{R}^k; \mathbb{R})$ (piecewise differentiable except at $g_i = 0$ or $h_j = 0$) maps from a sample $x$ and parameter $y$ and outputs the total constraint violation as a scalar value. In fact, each constraint term can be further customized with different weights that allow different treatments. The violation value is expected to decrease during the training process of diffusion models. We introduce the newly proposed loss function and the corresponding training process as follows.

**DiffuSolve+ Training.** For the diffusion process, we try to generate samples from solution set $x_0 \in \mathcal{X}$. By reformulating the forward sampling process as in Eq. (3), we have $x_0 = \frac{x_k - \sqrt{1-\bar{\alpha}_k}\varepsilon}{\sqrt{\bar{\alpha}_k}}, \varepsilon \sim \mathcal{N}(0, \mathbf{I})$. The noise $\varepsilon$ is approximated as the neural-network parameterized model $\varepsilon_\theta(x_k, k, y)$ for condition $y$ at timestep $k$. Replacing it into previous formula, the clean sample at timestep $k = 0$ is predicted as $\hat{x}_0 = \frac{x_k(x_0, \varepsilon) - \sqrt{1-\bar{\alpha}_k}\varepsilon_\theta(x_k, k, y)}{\sqrt{\bar{\alpha}_k}}$. The benefits of direct $\hat{x}_0$ prediction is that it does not need iterative diffusion sampling thus avoids gradient backpropagation through multi-step sampling. For constrained diffusion generation with `DiffuSolve+`, we introduce a constraint violation loss $\mathcal{L}_{\text{vio}}$ on the predicted $\hat{x}_0$ from $x_k, \forall k \in [K]$:

$$\mathcal{L}_{\text{vio}} = \mathbb{E}_{(x_0, y) \sim \mathcal{X} \times \mathcal{Y}, k \in [K], \varepsilon \sim \mathcal{N}(0, \mathbf{I})} \left[ \frac{1}{k} V \left( \text{clip}\left(\hat{x}_0(x_k, \varepsilon_\theta), x_0^{\min}, x_0^{\max}\right), y \right) \right] \tag{7}$$

where the clip function preserves the range of $\hat{x}_0$ to be within the lower $x_0^{\min}$ and upper bound $x_0^{\max}$ for $x_0 \in \mathcal{X}$, ensuring the valid calculation of violation function $V(\cdot, \cdot)$. The reason for the clipping is that at the early phase of the training, the parameterized $\varepsilon_\theta$ as an approximation of $\varepsilon$ is not accurate thus the predicted $\hat{x}_0$ may not lie on the original range of the data $x_0$. The scheduling factor $\frac{1}{k}$ regulates the scale of the violation function $V$ for different $x_k, k \in [K]$, since the predicted $\hat{x}_0$ will be more noisy when $k$ is large and the estimation of violation function can be less reliable. This may not be a theoretically principled choice but is found effective and convenient in our experiments.

For the violation loss $\mathcal{L}_{\text{vio}}$, we always let the condition variable $y$ appear in the noise prediction $\varepsilon_\theta(x_k, k, y)$ without masking it out, contrary to the probabilistic masking in the original training loss $\mathcal{L}_{\text{diff}}$. An intuitive interpretation of the additional loss term is that it guides the diffusion model training with the gradients of minimizing the violation value for each denoising step.

Finally, our proposed constrained diffusion training loss is a combination of the original diffusion model training loss in `DiffuSolve` as Eq. 5 and the above violation loss:

$$\mathcal{L} = \mathcal{L}_{\text{diff}} + \lambda \mathcal{L}_{\text{vio}} \tag{8}$$

where $\lambda$ is a hyperparameter for adjusting the strength of $\mathcal{L}_{\text{vio}}$. As the training proceeds, the original loss function $\mathcal{L}_{\text{diff}}$ will encourage $\widehat{x}_0$ to be close to true $x_0$, and $\mathcal{L}_{\text{vio}}$ further enforces the constraint satisfaction of the generated samples.

**Conditional Diffusion Sampling.** For sampling from trained diffusion models, we adopt guided sampling with classifier-free guidance as introduced in Sec. 3. The guided prediction noise with guidance weight $c$ is:

$$\widehat{\varepsilon}_\theta \leftarrow c \cdot \varepsilon_\theta(x_k, k, y) + (1 - c) \cdot \varepsilon_\theta(x_k, k, \varnothing) \tag{9}$$

Then we plug $\widehat{\varepsilon}_\theta$ into $\varepsilon_\theta$ in the sampling process (Ho et al., 2020; Ho & Salimans, 2022), and through iterative sampling $x_{k-1}$ from $x_k (k = K, \ldots, 1)$ with initial random noise $x_K \sim \mathcal{N}(0, \mathbf{I})$, it produces the generated samples $x_0$. The diffusion sample serves as the predicted initial guess in Sec. 4.1 for a new task: $\tilde{x}^*_{\text{new}} := x_0$.

# 5 EXPERIMENT

In this section, we first evaluate `DiffuSolve` with a vanilla diffusion model (DM) on three high-dimensional and highly nonlinear trajectory optimization tasks. Secondly, we present the results of `DiffuSolve+` with our novel constrained diffusion model (CDM). As a standalone method, the CDM reduces problem constraint violation of the samples compared to DM. Within `DiffuSolve+`, CDM helps further improve the efficiency of obtaining locally optimal and feasible solutions compared to `DiffuSolve`.

The experiments mainly involve three stages: 1. Use an NLP solver to collect solutions on each task domain; 2. Train the data-driven models to fit the solution dataset; 3. For unseen new tasks within each domain, sample the initial guesses from each model that warm-start the solver to derive final solutions. The details of data collection and model training refer to Appendix A.

## 5.1 TASK SETUP

Our experiments are conducted on three complex non-convex trajectory optimization tasks as illustrated in Figure 1. These tasks span diverse domains and present unique challenges: (i). *Two-Car Reach-Avoid*: Multi-agent game-theoretical problem is difficult to scale and features multiple equilibria. (ii). *Quadrotor Navigation*: High-dimensional nonlinear dynamical systems lead to computational inefficiency. (iii). *Cislunar Transfer*: The highly nonlinear chaotic dynamical system makes spacecraft trajectory optimization time-consuming with numerous local optima. The detailed dynamical settings for each task are as follows. The numerical settings can be seen in Appendix A.

**Two-Car Reach-Avoid.** This is a two-player game inspired by (Chen et al., 2018). Each car tries to find a minimum-time path to reach the goal while avoiding collision with another car and obstacles, shown in Figure 1. Specifically, both $car_1$ and $car_2$ follows the same dynamics as follows:

$$\dot{p}_x = v \cos\theta, \quad \dot{p}_y = v \sin\theta, \quad \dot{v} = a, \quad \dot{\theta} = \omega,$$

where $(p_x, p_y, v, \theta)$ denotes the state of each car. $p_x, p_y$ are the position, $v$ is the speed, and $\theta$ is the orientation of the car. Each car has two-dimensional controls $u^1 = (a^1, \omega^1), u^2 = (a^2, \omega^2)$, where $a \in [-1, 1]$ is the acceleration and $\omega \in [-1, 1]$ is the angular speed.

**Quadrotor Navigation.** This a navigation problem of a 10D quadrotor, inspired by (Herbert et al., 2017). The quadrotor needs to find the minimum time to reach the goal position while avoiding obstacles shown in Figure 1, where the goal and obstacles are randomly sampled. The quadrotor's dynamics is defined as follows:

$$\dot{x} = v_x, \quad \dot{v}_x = g \tan\theta_x, \quad \dot{\theta}_x = -d_1\theta_x + \omega_x, \quad \dot{\omega}_x = -d_0\theta_x + n_0 a_x, \quad \dot{z} = v_z,$$

$$\dot{y} = v_y, \quad \dot{v}_y = g \tan\theta_y, \quad \dot{\theta}_y = -d_1\theta_y + \omega_y, \quad \dot{\omega}_y = -d_0\theta_y + n_0 a_y, \quad \dot{v}_z = K_T a_z - g \tag{10}$$

where $x, y, z$ are the position of the quadrotor. $\theta_x, \theta_y$ are the pitch and roll angle, and $\omega_x, \omega_y$ are the corresponding rates. $v_x, v_y, v_z$ are the speed of the quadrotor. $d_0, d_1, K_T, n_0$ are system parameters, and $g = 9.81$. The controls are $u = (a_x, a_y, a_z)$.

**Cislunar Transfer.** In this task, we consider a minimum-fuel *cislunar low-thrust transfer* with the dynamical model to be a Circular Restricted Three-Body Problem (CR3BP) (Koon et al., 2000). As shown in Figure 1, the example trajectory (orange) starts from the end of a geostationary transfer spiral (blue) and ends at an arc of the invariant manifold (black). Assuming the mass of the spacecraft is negligible, the CR3BP describes the equation of motion of a spacecraft under the gravitational force from the Earth and moon. Let $m_1$ be the mass of the Earth and $m_2$ be the mass of the moon, and $\mu = m_2/(m_1 + m_2)$, the dynamics of the spacecraft in the CR3BP is as follows:

$$\ddot{x} - 2\dot{y} = -\bar{U}_x, \quad \ddot{y} + 2\dot{x} = -\bar{U}_y, \quad \ddot{z} = -\bar{U}_z,$$

where $\bar{U}(r_1(x, y, z), r_2(x, y, z)) = -\frac{1}{2}\left((1-\mu)r_1^2 + \mu r_2^2\right) - \frac{1-\mu}{r_1} - \frac{\mu}{r_2}$ is the effective gravitational potential, $r_1 = \sqrt{(x+\mu)^2 + y^2 + z^2}$ and $r_2 = \sqrt{(x-(1-\mu))^2 + y^2 + z^2}$ are the distance from the spacecraft to the sun and moon, respectively. This is a chaotic dynamical system.

## 5.2 EXPERIMENT SETUP

**Numerical Solvers.** We assume fully known dynamics and environment for the open-loop trajectory optimization problem. For all three tasks, we formulate and solve the optimization problem with Sparse Nonlinear OPTimizer (SNOPT) (Gill et al., 2005) as the solver $\pi$, which uses the SQP method and supports warm-starting. For the *cislunar transfer*, we adopt *pydylan*, a Python interface of the Dynamically Leveraged Automated (N) Multibody Trajectory Optimization (DyLAN) (Beeson et al., 2022) to formulate the CR3BP dynamics in addition to SNOPT. The detailed problem formulation and solving for each task are included in Appendix A.

**Baselines.** For comparison with the proposed `DiffuSolve`, we introduce the following baselines:

- **Uniform**: This method solely uses an NLP solver with uniformly sampled initial guesses $x \in \mathcal{X}$ for the non-convex trajectory optimization problem. It solves problems from scratch without leveraging any problem-specific prior information.
- **Optimal Uniform**: This method uniformly samples from collected locally optimal solutions $\{x^*\}$ as initial guesses for the NLP solver. It establishes a performance lower bound for data-driven methods since without learning it cannot generalize to new problems.
- **CVAE_LSTM**: This method (Li et al., 2023) combines the convolutional variational auto-encoder (CVAE) with long short-term memory (LSTM) to learn the conditional distribution of locally optimal solutions. Specifically, a CVAE with Gaussian Mixture Model (GMM) prior is first used to sample the non-control variables, e.g. time or mass variables, and then an LSTM is adapted to sample the control sequence.

Although there exist graph-based methods for path planning, such as the Rapidly-exploring Random Trees (RRT) (LaValle, 1998), A* (Hart et al., 1968), etc., they do not aim to produce optimal solutions, nor handle the case of nonlinear dynamical systems in complex environmentally constrained problems well. Thus, the comparison against them is not provided in our experiments.

**Diversity Measure.** We adopt the Vendi Score (Friedman & Dieng, 2022) as a quantitative metric for measuring the diversity of the model samples. The similarity kernel we choose is $k(x, y) = e^{-||x-y||}$.

## 5.3 DIFFUSOLVE RESULTS

In this section, we evaluate `DiffuSolve`'s performance with a vanilla diffusion model (DM) against baseline methods, focusing on solution optimality, feasibility, diversity, and computational efficiency.

### 5.3.1 OPTIMALITY AND FEASIBILITY

We generate 600 samples from each method and use them as warm starts for the NLP solver. In Table 1, we present the percentage (ratio) of locally optimal and feasible solutions over 600 samples

| Task | Method | | | | |
|------|--------|--|--|--|--|
| | DiffuSolve+ | DiffuSolve | CVAE_LSTM | Uniform | Optimal Uniform |
| Two-car | **94.83** | 94.50 | 63.17 | 64.17 | 83.00 |
| Quadrotor | **100.00** | 99.83 | 99.67 | 99.17 | 98.50 |
| Cislunar | – | **54.00** | 29.17 | 17.50 | 23.33 |

*Note:* All values are for '+ Solver' (measured on solver outputs with initial guesses from model samples). '- Solver' values (measured on raw model samples) were 0% for all cases and are omitted for brevity.

Table 1: Percentage (%) (ratio) of locally optimal and feasible solutions over 600 samples.

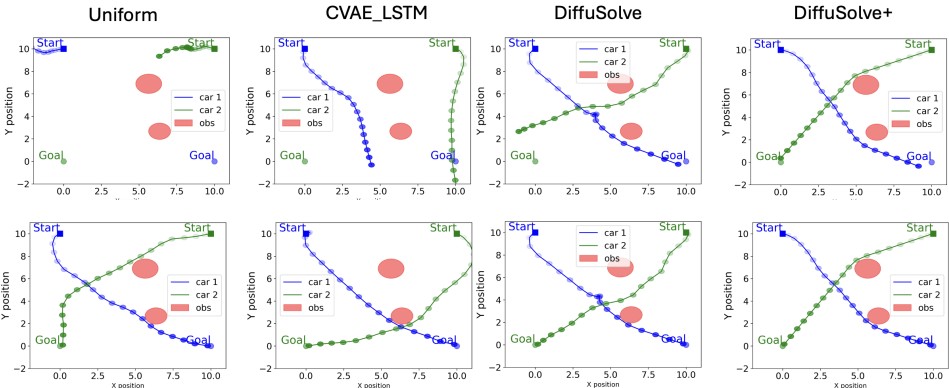

Figure 3: In *two-car reach-avoid* problem: First row: raw samples (no solver); Second row: corresponding solver derived locally optimal and feasible solutions.

which are fine-tuned by the NLP solver. Note that before finetuning, this ratio is zero for all methods' raw samples. The optimality and feasibility are verified by checking the KKT condition and the constraint violation based on the task setup in Sec. 5.1. As shown in Table 1, no method is able to directly generate locally optimal and feasible solutions. When using the samples as warm starts for the NLP solver, our proposed `DiffuSolve` obtains a greater number of locally optimal and feasible solutions than any non-diffusion methods.

We also visualize the sampled trajectories and the corresponding solver-derived trajectories in Figure 3. For the Uniform and CVAE_LSTM method, the generated trajectories are far from feasible, thus many of them cannot serve as good initial guesses for the NLP solver. `DiffuSolve` already generates near-feasible trajectories and thus makes NLP solver easier to derive locally optimal and feasible solutions.

### 5.3.2 COMPUTATIONAL EFFICIENCY

For the locally optimal and feasible solutions obtained in Table 1, Table 2 presents the computational time statistics, involving both sampling time from the model and the NLP solver time. In Table 2, the `DiffuSolve` with a vanilla DM is 2 × to 11 × faster than the uniform method based on the top 25%-quantile of the computational time for deriving the locally optimal and feasible solutions. Also in the histogram plots in Figure 5, `DiffuSolve` obtains more solutions than baseline methods within a short period of time. These results coincide with Figure 3 that since the `DiffuSolve` generates samples that are close to local optima, it takes a shorter time for the NLP solver to converge.

Thus, it motivates us to set a proper cut-off time for the solver in the future, which can allow more initial guesses to be fed. This could further improve the efficiency of `DiffuSolve` to derive locally optimal and feasible solutions.

### 5.3.3 DIVERSITY

In this paper, our proposed `DiffuSolve` aims to learn the solution distribution of the non-convex problem, instead of predicting a single solution. For example, for the *two-car reach-avoid* problem, there are multiple equilibrium points that are locally optimal and feasible. In Figure 4, we present

| TASK | METHOD | SOLVING TIME | | | VENDI SCORE |
|---|---|---|---|---|---|
| | | MEAN($\pm$STD) | 25%-QUANTILE | MEDIAN | |
| TWO-CAR | DIFFUSOLVE+ (OURS) | **17.86 $\pm$ 14.28** | **7.68** | **13.83** | **3873.60** |
| | DIFFUSOLVE (OURS) | 18.82 $\pm$ 14.33 | 8.77 | 15.61 | 3574.66 |
| | CVAE_LSTM | 36.24 $\pm$ 20.55 | 19.45 | 33.26 | 1162.72 |
| | UNIFORM | 46.17 $\pm$ 20.63 | 29.62 | 43.54 | **5256.75** |
| | OPTIMAL UNIFORM | 25.15 $\pm$ 19.75 | 10.82 | 20.89 | 3708.58 |
| QUADROTOR | DIFFUSOLVE+ (OURS) | **6.65 $\pm$ 3.55** | **4.47** | **6.05** | 5458.23 |
| | DIFFUSOLVE (OURS) | 7.30 $\pm$ 4.79 | 4.78 | 6.62 | 5428.63 |
| | CVAE_LSTM | 9.05 $\pm$ 8.30 | 5.00 | 6.74 | 309.18 |
| | UNIFORM | 16.87 $\pm$ 13.60 | 9.28 | 12.48 | **5985.10** |
| | OPTIMAL UNIFORM | 13.94 $\pm$ 12.08 | 7.66 | 10.13 | **5575.83** |
| CISLUNAR | DIFFUSOLVE (OURS) | **50.81 $\pm$ 70.30** | **9.97** | **26.68** | 5999.96 |
| | CVAE_LSTM | 141.31 $\pm$ 125.60 | 38.41 | 106.48 | 1403.38 |
| | UNIFORM | 199.34 $\pm$ 116.59 | 114.02 | 174.35 | **6000.00** |
| | OPTIMAL UNIFORM | 177.35 $\pm$ 123.22 | 70.03 | 153.97 | 5918.08 |

Table 2: Computational time (sampling + solving) statistics and Vendi Score.

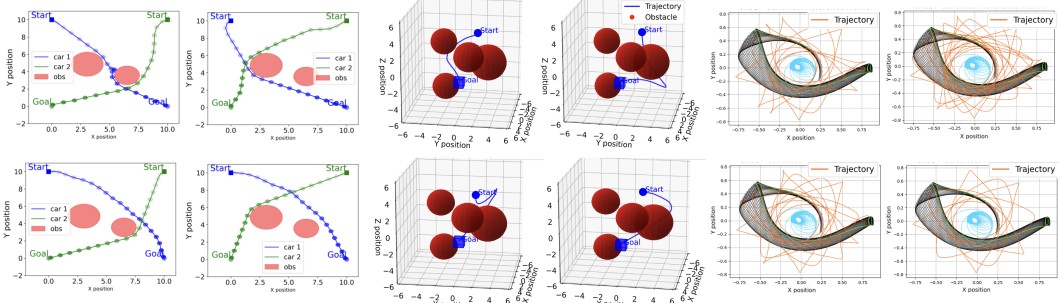

Figure 4: Diverse trajectory solutions for *two-car reach-avoid*, *quadrotor navigation* and *cislunar transfer* problem with `DiffuSolve` method, given the same conditional input.

4 qualitatively different solutions with `DiffuSolve` for each task, showing its ability to generate diverse solutions even within a shorter period of time.

The Vendi Score is also evaluated on the samples from each method in Table 2. While the Uniform method certainly generates the most diverse samples, the `DiffuSolve` is comparable to Optimal Uniform and outperforms CVAE_LSTM in sample diversity. Especially for the *quadrotor navigation*, though the solving time for CVAE_LSTM is competitive in Table 2, the Vendi Score shows that its generated solutions are not diverse at all.

### 5.4 DIFFUSOLVE+ RESULTS

In this section, we present the results of `DiffuSolve+` on the *quadrotor navigation* and *two-car reach-avoid* task. The proposed CDM is evaluated both as a standalone method for reducing constraint violations and within the `DiffuSolve+` framework for improving computational efficiency.

Since the current *cislunar transfer* task originally adopted from (Beeson et al., 2022) uses an adaptive-stepsize integrator for the dynamics constraint, it requires extra effort to incorporate it into the neural network and is beyond our scope. We leave the CDM for this task as future work.

#### 5.4.1 CONSTRAINT VIOLATION

As a standalone method without the NLP solver, our proposed CDM from Sec. 4.2 has fewer constraint violations compared to the vanilla DM and CVAE_LSTM, displayed in Table 3. For the *quadrotor navigation*, the CDM is able to directly generate feasible solutions more than any other methods. Also in Figure 3 for the *two-car reach-avoid* problem, the CDM generates a trajectory that is the closest to the locally optimal and feasible solutions - only car 1 cannot reach the goal but it is already near. It will be easy for the NLP solver to close this gap within `DiffuSolve+`.

| TASK | METHOD | CONSTRAINT VIOLATION (NO SOLVER) | | | FEASIBLE RATIO ‰ |
|---|---|---|---|---|---|
| | | MEAN($\pm$STD) | 25%-QUANTILE | MEDIAN | (NO SOLVER) |
| TWO-CAR | DIFFUSOLVE+ | **3.00 $\pm$ 15.36** | **0.73** | **1.44** | 0 |
| | DIFFUSOLVE | 10.99 $\pm$ 35.59 | 1.98 | 4.52 | 0 |
| | CVAE_LSTM | 282.72 $\pm$ 186.40 | 147.09 | 292.90 | 0 |
| QUADROTOR | DIFFUSOLVE+ | **3.06 $\pm$ 7.47** | **0.10** | **0.38** | **81.8** |
| | DIFFUSOLVE | 3.85 $\pm$ 8.52 | 0.16 | 0.63 | 49.6 |
| | CVAE_LSTM | 20.19 $\pm$ 22.32 | 3.36 | 12.96 | 16.3 |

Table 3: Left: The constraint violation of the model samples. Right: The ratio of feasible solutions. Both are measured directly on the model samples without the NLP solver. Here `DiffuSolve+`: constrained diffusion model, `DiffuSolve`: vanilla diffusion model

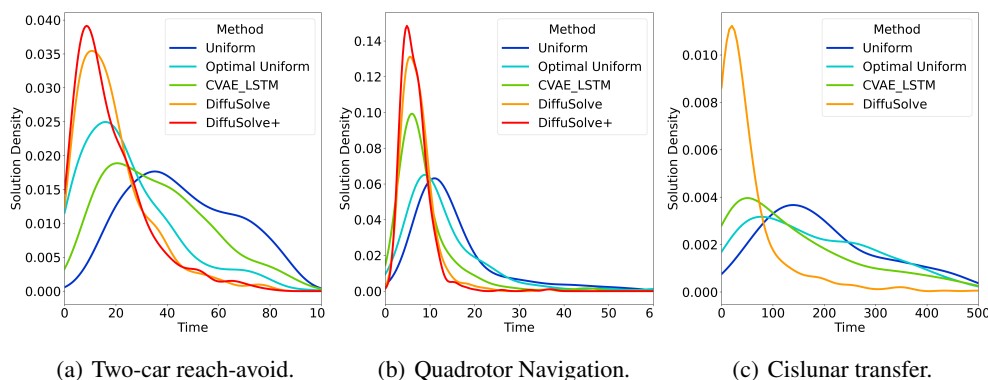

(a) Two-car reach-avoid.  (b) Quadrotor Navigation.  (c) Cislunar transfer.

Figure 5: The histogram of computational time (including model sampling time and solver time) for different methods to find locally optimal and feasible solutions.

### 5.4.2 COMPUTATIONAL EFFICIENCY

For `DiffuSolve+` with the proposed CDM, the computational efficiency is further improved as shown in Table 2. The intuition is clear from Figure 3: when the `DiffuSolve+` sample is closer to the local optima, it takes the NLP solver an even shorter time to derive the final solution. We also present the histogram of the computational time in Figure 5. Here we visualize the density of locally optimal and feasible solutions obtained within a time range, where `DiffuSolve+` in red is able to obtain more solutions than `DiffuSolve` within a short period of time from 600 initial guesses.

## 6 CONCLUSION AND DISCUSSION

This paper presents `DiffuSolve`: a general diffusion model-based solver for non-convex trajectory optimization. It can generate locally optimal, feasible, and diverse solutions with high computational efficiency. We also propose `DiffuSolve+` which includes a novel constrained diffusion model. It has fewer constraint violations as a standalone method and further improves the computational efficiency when integrated with `DiffuSolve+`. It is also a general acceleration framework that has the potential to solve optimization problems in other areas beyond trajectory optimization such as finance, computational chemistry, etc.

Limitations also exist for the current paper. We don't spend extra effort to optimize the diffusion model's sampling speed. As a result, the current `DiffusSolve` and CDM may not be able to achieve real-time optimization for high-dimensional systems or highly dynamic environments. This can be improved with existing acceleration techniques such as stride sampling, DDIM (Song et al., 2020a), etc. In addition, although most constraint functions in trajectory optimization are differentiable (almost everywhere), the current CDM does not work for non-differentiable violation functions. For non-differentiable constraints, transforming them into differentiable functions or applying approximation with appropriate bump functions or neural networks is possible.

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

## A  IMPLEMENTATION AND TASK DETAILS

### A.1  MODEL DETAILS

For both vanilla and constrained diffusion models (DMs), we apply a UNet structure with three hidden layers of 512, 512, and 1024 neurons. We also use a fully connected layer of 256 and 512 neurons to embed the conditional input $y$. The sampling step is set to be 500. Both DMs are trained with 200 epochs using the Adam optimizer (Kingma & Ba, 2014). The constrained and unconstrained DMs usually take 14 hours and 9 hours respectively as well as 20 GB VRAM to train on one NVIDIA A100 GPU, for every 100k training data with batch size of 512.

### A.2  TWO CAR REACH-AVOID

**Problem formulation and solving.**  In this task $\mathcal{P}_y$, we aim to minimize the time for the two cars to reach each own goal while avoiding colliding with the other car and obstacles. We fixed the start $p_{\text{start}}$ and goal $p_{\text{goal}}$ position for $car_1$ to be $(0, 10)$ and $(10, 0)$, and for $car_2$ to be $(10, 0)$ and $(0, 0)$. The conditional parameters $y \in \mathbb{R}^6$ include the two obstacle positions $p_{\text{obs}}$ randomly sampled from $[2.0, 8.0]$, and the obstacle radius $r_{\text{obs}}$ randomly sampled from $[0.5, 1.5]$. We formulate this trajectory optimization problem with a forward shooting method, with the discretized time step $T = 20$. The variable to optimize is $x = (t_{\text{final}}, u_1^1, u_2^1, ..., u_T^1, u_1^2, u_2^2, ..., u_T^2) \in \mathbb{R}^{41}$, where $u_i^1$ and $u_i^2$ are controls for two cars, respectively. The problem parameter is $y = (p_{\text{obs}}, r_{\text{obs}})$. The objective is defined as $J(x; y) = t_{\text{final}}$. The constraint $g$ includes the goal-reaching constraints at $t_{\text{final}}$, and the collision avoidance for obstacles and between each car for all time. We use the 4-th order Runge-Kutta (RK4) method to integrate the dynamics and formulate the constraints.

**Data collection and model training.**  In this task, we collect locally optimal solutions $x^*$ from 3k different $y$, each with 100 uniformly sampled initial guesses using SNOPT (Gill et al., 2005). We filter out those solutions whose objective is higher than 12 s and use the remaining 114k solutions as the training data. Within this 114k data, 10% of the data are used as the validation set. The whole data collection process takes around 8 hours with 200 AMD EPYC 9654 CPU cores.

We train each of our proposed constrained and unconstrained DMs and baseline models with 3 different random seeds and test the warm-starting performance with 600 initial guesses sampled from 3 random seeds across 20 unseen obstacle settings, i.e. unseen $y$ values.

### A.3  QUADROTOR NAVIGATION

**Problem formulation and solving.**  In this task $\mathcal{P}_y$, we aim to minimize the time for the quadrotor to reach the goal while avoiding the obstacles. The start position is fixed at $(-12, 0, 0)$, and the reference goal position is $(12, 0, 0)$. The conditional parameter $y$ includes the perturbation $d_{\text{goal}}$ randomly chosen from $[-2, 2]$ that is added to the goal position, 4 obstacle positions $p_{\text{obs}}$ randomly sampled between the start and goal, and the obstacle radius $r_{\text{obs}}$ randomly sampled from $[1.5, 3.5]$. We discretize this problem into $T$ timesteps where $T = 80$. The variable to optimize is $x = (t_{\text{final}}, u_1^1, .., u_T^1, u_1^2, ...u_T^2, u_1^3, ...u_T^3) \in \mathbb{R}^{241}$, and the conditional parameter is $y = (d_{\text{goal}}, p_{\text{obs}}, r_{\text{obs}}) \in \mathbb{R}^{17}$. The objective is defined as $J(x; y) = t_{\text{final}}$. The constraint $g$ includes the goal-reaching constraints at $t_{\text{final}}$ and the collision avoidance for obstacles at all time. We use the 4-th order Runge-Kutta (RK4) method to integrate the dynamics and formulate the constraints.

**Data collection and model training.**  In this task, to collect the training data we sample 4000 different $y$, each with 50 uniformly sampled initial guesses. Then we use the SNOPT (Gill et al., 2005) solver for each problem instance to collect locally optimal solutions. Finally, we filter out those solutions with objective values greater than 4.57 s and use the remaining 179k data as training data. Within this 179k data, 10% of the data are used as the validation set. The whole data collection process takes around 20 hours with 200 AMD EPYC 9654 CPU cores.

We train each of our proposed constrained and unconstrained DMs and baseline models with 3 different random seeds. The models are tested for the warm-starting performance with 600 initial guesses from 3 random seeds across 20 unseen locations of the goal and obstacles, i.e. unseen $y$ values.

## A.4 Cislunar Transfer

**Problem formulation and solving.** In this problem $\mathcal{P}_y$, as shown in Fig. 1, the spacecraft is planned to start from a Geostationary Transfer Spiral (blue) and reach a stable manifold arc (green) of a Halo orbit near $L_1$ Lagrange point (green). A candidate trajectory solution is plotted in orange. The CR3BP is a chaotic system, where a smaller perturbation will lead to a significantly different final trajectory. Therefore, trajectory optimization of this problem contains many local optima. We choose the conditional parameter $y \in [0.1, 1.0]$ Newton to be the maximum allowable thrust of the spacecraft.

We use the forward-backward shooting method to formulate the trajectory optimization problem in Eq. (1). For the spacecraft, we set the constant specific impulse (CSI) $I_{sp} = 1000$ s. The initial dry mass is 300 kg and the initial fuel mass is 700 kg. We choose $T$ discretized control segments where $T = 20$ and the control $u$ are uniformly applied on each segment. The variable $x$ to optimize is as follows:

$$x = (t_{\text{burn}}, t_{\text{coast}}^{\text{initial}}, t_{\text{coast}}^{\text{final}}, m_f, u_1, u_2, ..., u_N) \in \mathbb{R}^{64} \tag{11}$$

where $t_{\text{burn}}, t_{\text{coast}}^{\text{initial}}, t_{\text{coast}}^{\text{final}}$ are the effective burning time for the engine, initial and final coast time, $m_f$ is the final mass, and $u_i$ is a three-dimensional thrust control of the spacecraft. The objective is defined as $J(x; y) = -m_f$, which is to minimize fuel expenditure. The constraint $g$ is the midpoint agreement when integrating dynamics from forward and backward. The problem parameter $y$ is the maximum allowable thrust for the spacecraft and is bounded by $[0.1, 1]$ Newton. To integrate the dynamics, we use the Runge-Kutta-Fehlberg 5-4th order (RK54) method (Fehlberg, 1969).

**Data collection and model training.** To collect the training data, we sample 12 different $y$ values from a grid in $[0.1, 1]$ Newton, and for each $y$ we collect 25k locally optimal solutions with $m_f \geqslant 415kg$ total amount of 300k data. The problem is solved using *pydylan*, a python interface of the Dynamically Leveraged Automated (N) Multibody Trajectory Optimization (DyLAN) (Beeson et al., 2022) and the solver SNOPT (Gill et al., 2005). Within this 300k data, 10% of the data are used as the validation set. The whole data collection process takes around 3 days with 500 Intel Skylake CPU cores.

We train the unconstrained DM and baseline models on 3 random seeds, and we test the warm-starting performance with 600 initial guesses sampled from 3 random seeds on $y = 0.15$ Newton, which is unseen in training data.

