# OpenReview forum: "DiffuSolve: Diffusion-Based Solver for Non-Convex Trajectory Optimization"
_ICLR.cc/2025/Conference — ICLR 2025 Conference Withdrawn Submission_

### Official Review · Reviewer_kcnh · 2024-10-29

**Soundness:** 2
**Presentation:** 3
**Contribution:** 2
**Rating:** 3
**Confidence:** 3

**Summary:**

The manuscript presents an algorithm for generating optimal trajectories for nonlinear dynamical systems. The presentation is mostly clear and the writing is OK. I have two major concerns:

1) Novelty

The manuscript suggests to use diffusion models to generate initial guesses for parametric nonlinear optimization problems. The initial guess is then refined by applying a nonlinear programming solver that also guarantees constraint satisfaction. Both building blocks (diffusion models) and NLP solvers have been used in the context of trajectory planning and combining the two, while certainly useful for solving an important applied problem, brings very little novelty/creativity.

As a result, I believe that the manuscript should maybe be better published at a robotics conference, rather than a machine learning venue.


2) Experiments

The experiments are somewhat limited (in particular, in view of the limited novelty). The planning problems that are considered have essentially about eight states (Two-Car Reach-Avoid), ten states (quadrotor navigation), six states (cislunar transfer), which is low dimensional. While nonconvex, the problems are also smooth and well-behaved (no contacts, friction, discontinuities, etc.). For context, similar planning problems are routinely solved in engineering, for more than twenty years (e.g. SNOPT dates back to 2005 and has been routinely used since then).
The diffusion model needs to be trained on a dataset that contains numerous solutions to the parametric program for different sets of parameters and initial conditions. The manuscript does not seem to consider the computational effort needed to assemble the training data when presenting the result, which would mean that the results are heavily skewed towards the authors' approach. In addition, from the presented Table (e.g. Table 2), I cannot see a 10-fold decrease in solution time, as claimed by the authors.


Summary: The manuscript is certainly interesting to read, however falls short of providing significant conceptual novelty or the solution of an open real-world problem in a convincing way.

**Strengths:**

see above.

**Weaknesses:**

see above.

**Questions:**

see above.

---

### Official Review · Reviewer_pZ3Z · 2024-10-31

**Soundness:** 3
**Presentation:** 3
**Contribution:** 3
**Rating:** 6
**Confidence:** 2

**Summary:**

This paper proposes a diffusion-based solver for solving the optimal trajectory in a nonlinear dynamical system with constraints. Traditional nonlinear programs (NLP) rely on the initial guess to find locally optimal and feasible solutions. The authors propose a framework called “DiffuSolve” that first samples high-quality solutions as warm starts and then uses the NLP solvers to refine them. The goal of the training process is to make the diffusion model generate diverse samples with minimal constraint violations and be able to generalize across different dynamics and constraints.

**Strengths:**

Planning optimally in a nonlinear dynamical system with constraints is a difficult but critical problem for many applications, such as robotics and autonomous driving. It is good to see that a diffusion model can help solve the problem more efficiently and with a higher success rate. The improvement in the experiment results looks significant.

**Weaknesses:**

My major concern is the novelty and the significance of “DiffuSolve” compared with previous works on diffusion-based trajectory generation with constraints. A straightforward idea is to use previous approaches to sample the warm-start trajectory and then use the NLP solver to fine-tune. In Related Works, the authors claim the difference is that this work focus on “high-dimensional”, “highly nonlinear”, and “diverse solutions”. Further, the design of the violation function (6) helps “DiffuSolve+” achieve better performances in problem instances with the above features. I do not see sufficient evidence on benchmark comparison to support these claims.

**Questions:**

To understand the contributions of different components in the "DiffuSolve" and "DiffuSolve+" frameworks, I suggest the authors try to generate the initial guesses with other existing approaches for a diffusion-based trajectory with constraints (in addition to "CVAE_LSTM"). Adding additional comparison benchmarks will help us understand whether most of the improvement comes from the proposed training method/loss functions or the idea of fine-tuning a diffusion model's guess by the NLP solver.

---

### Official Review · Reviewer_Fhzj · 2024-11-01

**Soundness:** 4
**Presentation:** 3
**Contribution:** 1
**Rating:** 3
**Confidence:** 4

**Summary:**

The paper presents two algorithms: DiffuSolve and DiffuSolve+. In DiffuSolve, a conditional diffusion model is first trained to learn $p(x^*|y)$ with a loss function derived from Ho & Salimans, 2022. An offline data generation process uses a nonlinear program solver, $\pi$, to collect locally optimal and feasible solutions $x^*$ for problem instances $P_y$ with uniformly sampled initial guesses. It manually filters out samples corresponding to suboptimal objectives. The diffusion model predicts diverse guesses $\tilde{x}^*_{new}$, which the NLP solver $\pi$ then uses to derive final solutions. Intuitively, this warm-starting approach with diverse samples helps find a global minimizer in the presence of multiple local minima, making it amenable to nonconvex optimization. The paper also develops DiffuSolve+ which ensures that no constraints are violated. For this, the algorithmic insight is to use a “closed-form” prediction of the sample $\hat{x}_0$, while minimizing a “clipped” objective which intuitively keeps predicted solutions within the feasible region.  The paper then runs three experiments: quadrotor navigation, two-car reach avoid, and cislunar transfer, and compares its algorithms against the baselines to demonstrate speed and efficacy results.

**Strengths:**

1) The paper is well-written - the technical details are explained clearly, and the experimental results are fairly presented.
2) The paper demonstrates good simulation results in three diverse settings.

**Weaknesses:**

1) The term "control transcription" (line 37) is not defined and is not used anywhere else in the paper. I think readers would appreciate it if you were to remove it if it is unnecessary, or if you were to explain it further, otherwise.

2) This is merely a pedantic point, but $\epsilon_{\theta}$ in lines 171 and 175 are being used for two things: one is a function of two variables, and the other is a function of three (not to mention there is an $\epsilon$ without a subscript!). I think this is fine, but maybe add a note saying “with abuse of notation”, or use a different variable for more clarity?

3) I just read "Model-Based Diffusion for Trajectory Optimization" (Pan, Yi, Chi, Qu, NeurIPS 24), which seems to have the same problem formulation and a similar algorithmic approach. Upon a closer look, I think there is a significant amount of overlap in the work.

4) While the literature review presented in the submission seems to touch on much of the existing work in the area, I'm not convinced that the submission contains any meaningful novelty. For instance, as the authors mention, it has been known that diffusion models are amenable to nonconvex optimization. In this sense, constructing a diffusion model and passing it to an optimizer is not novel, per se, which makes me question the contribution of DiffuSolve. On the other hand, DiffuSolve+ presents a nontrivial improvement with the clipping function to prevent violations; however, I wonder if there isn't already existing work in the literature to address preventing the infeasibility of solutions... In this regard, it would have been good to know the limitations of existing works in this area, which the literature review fails to highlight. For instance, the above reference (Pan et. al. 2024) solves some related problems in non-convex high-dimensional trajectory optimization and further references many other works, specifically on non-convex non-smooth high-dimensional trajectory optimization problems.

**Questions:**

1) In line 272, how are values for $\lambda$ determined?
2) I have also raised some questions in the weaknesses section. I would certainly appreciate a response to it!
3) All the current simulation results are being tested against "CVAE_LSTM" and some uniform samples. The paper doesn't mention, but is "CVAE_LSTM" the best-known baseline comparison? The experimental results might seem more meaningful if they were more robustly tested against a larger variety of algorithms from the literature...

---

### Official Review · Reviewer_Vt3r · 2024-11-03

**Soundness:** 2
**Presentation:** 3
**Contribution:** 2
**Rating:** 3
**Confidence:** 2

**Summary:**

This paper proposes to use diffusion models (DMs) to enhance the nonlinear program (NLP) solvers for trajectory optimization. The authors first collect local-optimal trajectories of related tasks to train a DM, then use the samples generated from the DM to warmup the NLP solver. They also derive a new loss function to enforce the DM to consider feasibility during sampling. The experiment results from 3 constrained trajectory optimization tasks demonstrate that the proposed DiffuSolve and Diffusolve+ generate better initial samples compared to non diffusion-based baselines, reducing the solving time of NLP solvers.

**Strengths:**

1. The paper is clear and well-written.

2. The use of diffusion models helps better warm-up the trajectory optimization solvers.

**Weaknesses:**

1. The authors basically directly incorporate DMs in the DiffuSolve pipeline, where the new proposed loss function seems contribute marginally when combined with NLP solvers. From Table 1, DiffuSolve+ produces only two additional feasible solutions in the first task and one more in the second task compared to the original DiffuSolve.

2. The demonstrated effectiveness of DiffuSolve relies on in-distribution, near-optimal data derived from tasks with similar parameters. It would be intersting to  assess the generalization capabilities of DiffuSolve when the distribution of training parameters differs from the distribution encountered during online testing.

**Questions:**

1. Are there any tasks that can illustrate the advantage of DiffuSolve+ over the DiffuSolve when combined with NLP solvers (maybe more safety critical applications)?

2. In the related work, the authors mention several works of diffusion-based trajectory generation with constraints. Why they are not included in the experiment baselines?

3. What value is assigned to the hyperparameter $\lambda$  during the experiments? An ablation study on this hyperparameter may help better understanding the contribution of the proposed loss function.

---

### Note · Authors · 2024-11-22

**Comment:**

We thank the reviewers for reviewing our paper. After careful consideration, we decided to withdraw our paper.

**Withdrawal Confirmation:**

I have read and agree with the venue's withdrawal policy on behalf of myself and my co-authors.